# PerFIT: Personalized Federated Instruction Tuning via Neural Architecture Search

## Abstract

Federated Instruction Tuning (FIT) has shown the ability to enable model instruction tuning among massive data owners without exposing privacy. However, it still faces two key challenges, i.e., data and resource heterogeneity. Due to the varying data distribution and preferences among data owners, FIT cannot adapt to the personalized data of individual owners. Moreover, clients with superior computational abilities have to compromise to maintain the same fine-tuning architecture as the weaker clients. Such a constraint prevents the powerful clients from having more trainable parameters for better fine-tuning performances. To address these issues uniformly, we propose a novel **Per**sonalized **F**ederated **I**nstruction **T**uning (**PerFIT**) framework based on architecture search. Specifically, PerFIT allows each client to search for a personalized architecture by expanding the trainable parameter space of the global model, pruning them, and obtaining personalized sparse patterns. We further propose personalized parameter-wise aggregation to facilitate flexible aggregation among clients with diverse sparse patterns. This procedure allows personalized instruction fine-tuning within the expanded parameter spaces, concurrently preserving the same number of trainable parameters as the vanilla state, thus introducing no extra resource burden. The evaluations with multiple LLMs on various instruction-following datasets demonstrate that our approach can achieve up to a $23\%$ decrease in personalized perplexity compared to the state-of-the-art FIT methods.

## 1 Introduction

The emergent abilities of Large Language Models (LLMs) (23) have presented the powerful capability of solving various language-related tasks, including reasoning, text generation, and question-answering. To obtain better-aligned LLMs that can precisely follow the instructions of humans, Instruction Tuning (IT) (26; 25) has been proposed and demonstrated essential effectiveness in enhancing the generalizability of the foundation LLMs to downstream tasks. Compared to the conventional Fine Tuning (FT) methods, IT incorporates the vanilla text with specific instructions paired with corresponding answers, thereby unlocking the existing abilities of LLMs.

Although IT is superior to traditional FT, the success of IT greatly relies on the variety, quality, and quantity of the training data. In addition, the increasing concerns about data privacy (7) and the expensive expenses of data collecting and cleaning jointly impede the obtaining of large amounts of valuable data. Worse still, the heterogeneity of private data fails to reflect the meaningful statistical property of the domain, resulting in the implantation of inevitable bias during IT. To overcome the aforementioned issues, Federated Instruction Tuning (FIT) (32; 29) was proposed as the explorations of the instruction-based optimization framework in Federated Learning (FL). The two frameworks seamlessly integrated Parameter-Efficient Fine-Tuning (PEFT) methods (9; 14), enhancing the feasibility of lightweight local fine-tuning processes. Moreover, they showed that FIT can leverage instruction-following data with guarantees of privacy and improve the performance of LLMs.

Despite the fact the privacy-guaranteed FIT framework based on PEFT methods can alleviate data heterogeneity and allow collaborative training, the preference for local data is not taken into consideration. Existing FIT method ignores resource heterogeneity since every client has to share the same structure of fine-tuning modules, potentially causing the waste of resources on clients with more powerful capabilities given that more trainable parameters offer better fine-tuning performance (1).

To address the challenges of handling local data and resource heterogeneity (11), we propose an adaptive personalized federated instruction tuning method to enable local clients to fully use their data and resources. Our method is motivated by the intrinsic connection between data heterogeneity and architecture heterogeneity, thereby authorizing each client to search for a personal IT architecture. Specifically, we adopt the efficient foresight pruning method based on the Taylor expansion of the loss to simplify the expensive Neural Architecture Search (NAS) (16) process. Benefiting from the data-guided pruning, each client has a personal sparse structure of the IT modules that fit the personalized local data. Furthermore, we propose a personalized aggregation mechanism that achieves parameter-wise aggregation across clients to enhance the information interactions. Our contributions are summarized as follows:

- We develop a novel personalized federated instruction tuning method by exploring diverse local fine-tuning architectures based on heterogeneous local data. Our approach can simultaneously enable collaborative learning among clients with heterogeneous resources.
- We propose a personalized parameter-wise aggregation strategy for the fine-tuned modules to promote information interaction across local clients with various architectures.
- We conduct comprehensive experiments on three well-known LLMs and four instruction-following datasets in both resource heterogeneity and homogeneity scenarios, which adequately show the effectiveness of our method.

## 2 RELATED WORK

**Federated Instruction Tuning of Large Language Models.** Existing LLMs have demonstrated substantial performance in deriving task-relevant answers by simply decorating the vanilla input with instructions. However, the fine-tuning process is still a promising option to achieve better results when confronting unexplored tasks (18). To preserve the advantages of instruction data and fine-tuning, instruction tuning was proposed as an essential approach to optimize the performance of LLMs. This method improves the efficacy of LLMs in handling diverse and complex tasks by fine-tuning them with human instructions and aligning them with real-world tasks (28). Previous work in this area focuses on two ways to generate instructions: i) prompts manually created by humans (27) and ii) instruction-following data auto-generated by machines (25). Despite the fact that the first method is expensive, the quality of instruction data manufactured with human effects is elevated due to the precise human annotation. The latter utilizes a self-instruct method based on open-sourced LLMs to auto-generate instruction data. Specifically, a powerful LLM is deployed to generate massive task-specific instruction data, which is subsequently leveraged to boost the alignment ability of another trainable LLM. However, due to the high value of collecting instruction data for various tasks, the owners of specific data are unlikely to share it with other competitors (29). Thus, the data cross-silo scenarios still exist. In addition, the heavy burdens brought by full-parameter fine-tuning weaken the feasibility of conducting fine-tuning on local clients. To tackle these problems, the FIT frameworks proposed by (32; 29) provide a lightweight solution based on the Low-Rank Adapter (LoRA) (9) to overcome the challenge brought by heterogeneous data, but the personalization aspects of local clients including data and resource heterogeneity (e.g., number of trainable parameters that clients can afford) are not taken into consideration. Therefore, we delve into the LoRA-based fine-tuning method and propose a personalized FIT method to address both challenges simultaneously.

**Personalized Federated Learning.** Personalized Federated Learning (PFL) focuses on training a client-specific model to achieve better performance on each local dataset instead of a global model to accommodate all client data uniformly. Specifically, the personalization of clients includes two major aspects: i) data heterogeneity (17) and ii) resource heterogeneity (12). The former indicates the differences in local data distributions and the latter shows the diversity in terms of memory consumption, computation abilities, communication overhead, etc. To address the data heterogeneity challenges, existing methods including (20) introduced regularization terms to guide the local objectives. To tackle the challenge of resource heterogeneity, (19) proposed to distinguish personalized models from a global model through a hypernetwork. (30) derives Federated Neural Network Search (FL-NAS) to obtain personalized architectures based on data and resource heterogeneity. FedSelect (21) iteratively grows subnetworks of local personalized with decreasing sparsity values. While the previously mentioned methods are effective from certain viewpoints, most focus on a singular aspect of personalization. Worse still, none of them are tailored for PFL on LLMs. To address the two

personalization issues in a one-shot manner, we propose to leverage the concepts of NAS to conduct a fine-grained LoRA architecture search based on local data, aiming to meet the resource and data heterogeneity needs simultaneously.

## 3 PRELIMINARIES

**Neural Architecture Search (NAS).** Given a loss function $\ell_i$ and model parameters $\theta_i(\mathcal{A})$ based on an architecture $\mathcal{A}_i$, we formulate the architecture search as the following optimization problem:

$$\arg\min_{\mathcal{A}_i} \ell_i(\theta_i(\mathcal{A}_i); \mathcal{D}_i) \ \ s.t. \ R_i(\mathcal{A}_i) \leq B_i, \ i = 1, 2, ..., n \,. \tag{1}$$

Here, $R_i$ and $B_i$ represent the resource consumption and the budget limitation of the $i^{th}$ client. The budget of the $i^{th}$ client can be energy consumption, computational cost, bandwidth requirement, etc., or a combination of these. In this paper, we use the number of trainable parameters to reflect budget constraints and utilize the NAS to explore a personal training architecture for every client based on the local heterogeneous data $\mathcal{D}_i$.

**Low-Rank Adapter.** Given the significant constraints on computational resources and communication bandwidth for local clients, we focus on the LoRA (9) method to formulate FIT architectures. LoRA achieves the update of fine-tuning by constraining the update of model parameters to maintain a low intrinsic rank. For a pre-trained LLM parameterized by $\theta_{init} \in \mathbb{R}^{d \times k}$, LoRA utilizes a low-rank decomposition $\mathbf{AB}$ to represent the update $\Delta\theta$ where $\mathbf{A} \in \mathbb{R}^{d \times r}$, $\mathbf{B} \in \mathbb{R}^{r \times k}$ and the rank $r \ll min(d, k)$. The pre-trained parameter $\theta$ remains fixed during the fine-tuning while $A$ and $B$ are optimized. The update of $\theta_{init}$ is formed as

$$\theta_{new}\mathbf{x} = \theta_{init}\mathbf{x} + \Delta\theta\mathbf{x} = \theta_{init}\mathbf{x} + \mathbf{AB}\mathbf{x},$$

where $\theta_{new} \in \mathbb{R}^{d \times k}$ denotes the new weight which is re-parameterized after completing the fine-tuning. Note that for mainstream decoder-only LLMs, $d$ equals $k$.

**Personalized Federated Learning.** The goal of PFL is to train a personalized model for each client collaboratively. Considering $n$ clients with private Non-IID dataset denoted as $\mathcal{D}_n = \{(\mathbf{x}_{n,j}, y_{n,j})\}_{j=1}^{N_n}$, we want to solve the problem below:

$$\arg\min_{\mathbf{\Delta\Theta}} \frac{1}{n} \sum_{i=1}^{n} \mathcal{L}_i(\theta_{init}, \mathbf{\Delta}\theta_i), \ \mathcal{L}_i(\theta_{init}, \mathbf{\Delta}\theta_i) = \frac{1}{N_n} \sum_{j}^{N_n} \ell_i(\mathbf{x}_{n,j}, y_{n,j}; \theta_{init}, \Delta\theta_i).$$

$\theta_{init}$ and $\Delta\theta_i$ represent the frozen and trainable parameters of the $i^{th}$ client, respectively. $\ell_i$ is the loss function for the $i^{th}$ client. $\mathcal{L}_i(\mathbf{\Delta}\theta_i)$ denotes the average loss across the local data. $\mathbf{\Delta\Theta} = \{\mathbf{\Delta}\theta_i\}_{i=1}^{n}$ represents the set of trainable parameters ($\mathbf{A}$ and $\mathbf{B}$) in LoRA-based fine-tuning.

## 4 METHODOLOGY

### 4.1 WORKFLOW OF PERFIT

Figure 1 shows the workflow of our method. It consists of the following four major steps. ①**Local Architecture Search**: Local clients search for their personalized sparse masks. Then, the personalized sparse masks are transmitted to the server. ②**Sparse Module Generation and Local Fine-tuning**: Local clients generate personalized LoRA modules and conduct local fine-tuning. ③**Personalized Module Aggregation**: Local clients transmit the sparse fine-tuned LoRA modules to the server. The server performs parameter-wise personalized aggregation. ④**Personalized Module Generation and Distribution**: The server generates personalized LoRA modules and distributes them to clients to initialize a new round of local fine-tuning based on the global module and personalized sparse masks. The backbone of the LLM is frozen during both searching and federated training processes. ① and ② are conducted locally. ③ and ④ are conducted on the central server. Algorithm 1 exhibits the process of NAS. Algorithm 2 shows the details of the overall workflow, where the "Federated Tuning" includes ②, ③ and ④. Algorithm 3 explains the ③. Note that ① and ③ are the major components and will be detailed in the next section.

### 4.2 IMPLEMENTATION DETAILS

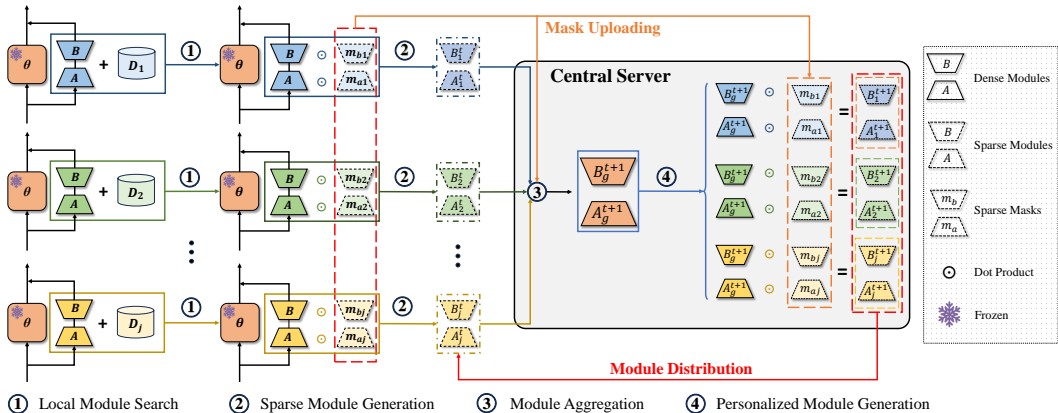

Figure 1: Workflow of our personalized federated instruction tuning approach.

**Local Architecture Search through Iterative Pruning.** For the $i^{th}$ client, we collaboratively search for the personalized architecture $\mathcal{A}_i$ that performs the best on the local dataset $\mathcal{D}_i$. Following Equation 1, the objective is defined as

$$\mathcal{A}_i = \arg\min_{\mathcal{A}} \mathcal{L}_i(\theta_i(\mathcal{A}), \mathcal{D}_i)$$

$$s.t. \ R_i(\mathcal{A}_i) \leq B_i, \mathcal{A}_i \neq \mathcal{A}_j \ for \ i \neq j,$$

where $\mathcal{L}_i(\cdot) = \sum_{i=1}^{n} p_i \mathcal{L}_i(\cdot)$ and $p_i = |N_n|/\sum_{i=1}^{n} |N_n|$ represents the ratio of the number of local data points to the number of overall data points. Given the budget of the number of trainable parameters $B_i$, our goal is

**Algorithm 1** NAS for LoRA modules

**Input** i) $\Delta\theta_0$, LoRA; ii) $T_p$, # of pruning epochs; iii) $n$, # of total clients; iv) $s$, sparsity.

1: **for** $i = 1, \ldots, n$ in parallel **do**
2:     **for** $t = 1, \ldots, T_p$ **do**
3:         Compute $I_{\Delta\theta_i}$ based on Equations 2-4;
4:         $\tau \leftarrow (1 - (1-s)^{t/T_p})$ percentile of $I_{\Delta\theta_i}$;
5:         $\mathbf{m}^i$ as $\mathbf{m}^i \leftarrow \mathbf{m}^i \odot (I_{\Delta\theta_i} < \tau)$;
6:     **end for**
7: **end for**
8: **Return** Sparse LoRA modules parameterized by $\Delta\theta_i \odot \mathbf{m}^i$

to find the LoRA architecture $\mathcal{A}_i$ which can achieve the best fine-tuning performance on local data $\mathcal{D}_i$. Due to the heavy burden of traditional NAS on LLMs, we perform the NAS on the LoRA module through foresight iterative pruning. Since pruning refers to the process from dense to sparse structure, we first replace the original LoRA module $\mathbf{A} \in \mathbb{R}^{d \times r}$ and $\mathbf{B} \in \mathbb{R}^{r \times d}$ with $\mathbf{A}_{de} \in \mathbb{R}^{d \times r/(1-s)}$ and $\mathbf{B}_{de} \in \mathbb{R}^{r/(1-s) \times d}$, respectively. Note that $s$ represents the sparsity and $0 < s < 1$. During pruning, we aim to remove the elements that have the least impact on the output of the model and reduce the number of parameters from $(d \times r/(1-s))\mathbf{X}$ to $(d \times r)\mathbf{X}$ by obtaining personalized mask $\mathbf{m}$ for each client. To estimate the importance of every element $\theta_i^j$ in $\mathbf{A}_d$ and $\mathbf{B}_d$ by ignoring higher order terms in Taylor expansion, we formulate the change of the loss as

$$I_{\Delta\theta_i^j} = \left| \frac{\partial \ell_i(\Delta\theta_i^j; \mathcal{D}_i)}{\partial \Delta\theta_i^j} \Delta\theta_i^j \right|, \tag{2}$$

where $\Delta\theta_i$ is represented by $\mathbf{A}_{de}^i \mathbf{B}_{de}^i$. Equation 2 shows the first-order estimation. Similarly, we can derive the parameter-wise second-order estimation as

$$I_{\Delta\theta_i^j} = \left| \Delta\theta_i^j H_{jj} \Delta\theta_i^j \right|. \tag{3}$$

$H$ represents the Hessian matrix and can be approximated by the Fisher information matrix to alleviate the computation overhead. For more generality, we integrate Equation 2 and 3 as the mixed metric, which is defined as follows:

$$I_{\Delta\theta_i^j} = \left| \frac{\partial \ell_i(\Delta\theta_i^j; \mathcal{D}_i)}{\partial \Delta\theta_i^j} \Delta\theta_i^j - \frac{1}{2} \Delta\theta_i^j H_{jj} \Delta\theta_i^j \right|. \tag{4}$$

Since $\mathcal{D}_i$ is the fine-tuning data that has never been used for the pre-training, the two terms in Equation 2 and 3 are not equal to zero, which shows that the proposed importance score is an ideal measurement of the importance of the architecture of the LoRA modules. The

overall process is described in Algorithm 1. In Line 3, we obtain the importance scores $I_{\Delta\theta_i}$. To avoid the potential layer collapse caused by over-confidence of one-shot pruning, we utilize an exponential decay schedule in Line 4 to determine the threshold value $\tau$ for pruning. After that, in Line 5, we mask the parameters whose importance scores are smaller than the threshold $\tau$ and preserve the rest. Different from the fine-grained NAS proposed by (13) that searches the parameters after training, we conduct the search before training to form a sparse training process.

**Symmetric Initialization.** Different from what was proposed in (9), we conduct the pruning-oriented NAS before starting training to avoid introducing expensive bi-level optimization. Nevertheless, due to the dependency of the importance measurement on the gradient, we need to carefully initialize the LoRA adapter to prevent *Measurement Vanishing*. Formally, the vanishing indicates that the values of importance scores are equal to zero, resulting in a diminished capability of the metric. Since the first and second-order terms rely on the gradient, we show that the vanishing happens without proper initialization. Based on the chain rule, the gradient of the $\mathbf{A}$ matrix in a LoRA module is defined as $\mathbf{g_A} = \frac{\partial \ell}{\partial o}\mathbf{B}$, where $\frac{\partial \ell}{\partial o}$ represents the gradient concerning the output of this layer. In vanilla LoRA configurations, the matrix $\mathbf{B}$ is initialized to all-zeros to avoid adding unexpected perturbations to

---

**Algorithm 2** Adaptive personalized FIT

**Input**: i) $\Delta\theta_0$, LoRA; ii) $T_p$, # of local pruning epochs; iii) $T_{tr}$, # of local fine-tuning epochs; iv) $k$; # of local clients in each round; v) $n$, # of total local clients; vi) $g_s$, a group of sparsity values.

1: **Local LoRA Module Search:**
2: **for** $i = 1, \ldots, n$ in parallel **do**
3:    Conduct Algorithm 1 based on the $i^{th}$ sparsity in $g_s$.
4: **end for**
5: **Federated Tuning:**
6: **for** $t = 1, \ldots, T_{tr}$ **do**
7:    $C_k \leftarrow$ Randomly sample $k$ clients from $n$ clients;
8:    $G_k \leftarrow$ Number of elements in $C_k$;
9:    **for** $j = 1, \ldots, G_k$ in parallel **do**
10:       Conduct $e$ epochs of local fine-tuning.
11:    **end for**
12:    Upload fine-tuned LoRA modules of clients in $C_k$;
13:    Conduct adaptive aggregation based on Algorithm 3;
14:    Dispatch personalized aggregated modules to clients in $C_k$.
15: **end for**
16: **Return** Personalized LoRA modules for each client.

---

the frozen backbone model. With such configurations, the gradient $\mathbf{g_A}$ is zero due to the state of $\mathbf{B}_{de}$, making the importance scores $I_{\mathbf{A}_{de}}$ all-zeros. Consequently, the pruning process only happens on the $\mathbf{A}_{de}$ matrix since the importance scores of $\mathbf{B}$ are always greater than 0. Therefore, such a problem will undermine the effectiveness of the pruning-oriented NAS process if we keep using the vanilla initialization. Accordingly, we follow the widely-used principle to symmetrically initialize $\mathbf{B}$ with the standard Gaussian and conduct the NAS process

$$\mathbf{A}_{de} \sim \mathcal{N}(0, 1/d), \ \mathbf{B}_{de} \sim \mathcal{N}(0, 1/d),$$

where $\mathcal{N}$ represents the Gaussian distribution.

**Personalized Aggregation.** To allow joint optimizations between local sparse patterns in a federated manner, we proposed a unified, personalized aggregation method for the LoRA modules. Formally, we can represent the pruned LoRA modules for the $i^{th}$ client as

$$\mathbf{A}^i_{T=0} = \mathbf{A}^i_{de,T=0} \odot \mathbf{m}^i_a, \ \mathbf{B}^i_{T=0} = \mathbf{B}^i_{de,T=0} \odot \mathbf{m}^i_b \tag{5}$$

where $\mathbf{m}^i_a$ and $\mathbf{m}^i_b$ denote the personalized mask matrices given the sparsity $s$. Since the pruning metric defined by Taylor expansion is dependent on the data $\mathcal{D}_i$, the obtained mask matrices vary across clients, i.e., $\mathbf{m}^i_a \neq \mathbf{m}^j_a$ and $\mathbf{m}^i_b \neq \mathbf{m}^j_b$. Intuitively, two personalized masks will not overlap if $\mathcal{D}_i$ is strictly heterogeneous to $\mathcal{D}_j$. For example, for a set of local LoRA-$\mathbf{A}$ modules $\{\mathbf{A}^1, \mathbf{A}^2, ..., \mathbf{A}^n\}$, we can mark each parameter $\mathbf{A}^z_{i,j}$ in $\mathbf{A}^{z\in n}$ with two states with respect to the parameter $\mathbf{A}^l_{i,j}$ in $\mathbf{A}^{l\in n}$: i) "*exclusive*"; and ii) "*shared*". Note that the states of each element can be conveniently obtained by the values of the corresponding sparse masks sent to the server from the beginning. Therefore, we formalize the personal aggregation matrix $\mathbf{\Gamma}^z$ for the $z^{th}$ client to realize the parameter-wise weighted aggregation. The new personalized LoRA for the $z^{th}$ is formed as

$$\mathbf{A}^z_{T+1} = \mathbf{m}^z_a \odot \sum_{z\in g_{id}} (\mathbf{A}^z_T \odot \mathbf{\Gamma}^z_A), \ \mathbf{B}^z_{T+1} = \mathbf{m}^z_b \odot \sum_{z\in g_{id}} (\mathbf{A}^z_T \odot \mathbf{\Gamma}^z_B),$$

where $\mathbf{\Gamma}^z_A$ and $\mathbf{\Gamma}^z_B$ represent the coefficient for LoRA-A and LoRA-B, respectively. $g_{id}$ is the indices that belong to the selected clients in round $T$. In Algorithm 3, Lines 2-3 explains the computation of the coefficient $\mathbf{\Gamma}^z_{i,j}$ for the element in position $(i, j)$. Note that with partial participation,

Since the states of each parameter implicitly form parameter-wise grouping, we only conduct the weighted aggregation within the same group. Based on the adaptive aggregation mechanism between sparse modules, we can further accommodate the fine-tuning process to resource heterogeneity scenarios. Since the main resource bottlenecks for local clients, including memory consumption and FLOPs, are inherently tied to the trainable parameters, we can adapt local module searches according to their maximum capability. Such targeted adaptation ensures optimal utilization of resources and empowers the overall performance. For-

---

**Algorithm 3** Generate $\mathbf{\Gamma}^z$ for the $z^{th}$ client.

---

**Input**: i) Index group $g_{id} = \{idx_1, idx_2, ..., idx_p\}$ of the selected clients ; ii) $\mathbf{M} = \{\mathbf{m}^{g_{id}^0}, \mathbf{m}^{g_{id}^1}, ..., \mathbf{m}^{g_{id}^p}\}$, local masks ; iii) $\{N^{g_{id}^0}, N^{g_{id}^1}, ..., N^{g_{id}^p}\}$, # data belongs to selected clients. iV) $i, j$, index of the element in the mask matrix; v) Index set $I = \{\}$.

**Generate $\mathbf{\Gamma}^z$:**

1: **for** all $\mathbf{m}_{i,j} \in \mathbf{M}^z$ in parallel **do**
2:      Perform $I \cup l$ when $\mathbf{m}_{i,j}^l = 1 \ \forall l \in g_{id}$;
3:      $\mathbf{\Gamma}_{i,j}^z = N^z / \sum_{k \in I} N^k$;
4: **end for**
5: **Return** Coefficient matrix $\mathbf{\Gamma}^z$ for the $z^{th}$ client.

---

mally, we first conduct Algorithm 1 based on a group of resource-specific sparsity levels $g_s = \{s_1, s_2, ..., s_m\}$ followed by applying Algorithm 3 to enable heterogeneous module aggregation.

**Computation and Time Complexity Analysis.** Similar to the work in (6) and (3), we applied 8-bit quantization on the frozen model and gradient-checkpoint methods to relieve the GPU memory burden when using the Adam optimizer to conduct local fine-tuning. Note that the computational resources required by NAS are less than those for fine-tuning. This is because the computational cost of NAS is as low as fine-tuning with the SGD optimizer, which is less complex than the Adam optimizer. Moreover, for each client in PerFIT, the pruning is performed only once at the first round of local training. As a result, the computational overhead of pruning in PerFIT is negligible in practice. For each client, we maintain a bitmap data structure to represent the mapping between client model parameters and their counterparts in the expanded space. Our aggregation operation described in Algorithm 3 has a time complexity of $O(MK)$. Here, M is the number of LoRA modules, and K is the number of selected clients in each FL communication round.

## 4.3 Convergence Analysis

We present the convergence analysis of our PerFIT method. Since our local NAS method is derived from iterative pruning and forms a static sparse pattern on parameter space, we establish the convergence property from the perspective of sparse training. We make the following assumptions.

**Assumption 1.** *(Coordinate-wise bounded gradient discrepancy). For any $\Delta\theta \in \mathbb{R}^{d \times r}$, there exists a constant $C \geq 0$ such that $\left\| \nabla \mathcal{L}_i(\Delta\theta) - \frac{1}{n} \sum_{j=1}^{n} \nabla \mathcal{L}_j(\Delta\theta) \right\|_\infty \leq G$.*

**Assumption 2.** *(Coordinate-wise bounded gradient). The local gradient of each client is bounded by the constant B such that $\|\nabla_{\Delta\theta} \mathcal{L}_i(\boldsymbol{w})\|_\infty \leq B$.*

**Assumption 3.** *(Bounded variance). The gradient $\boldsymbol{g}_{i,t,\tau}(\Delta\theta) := \nabla\ell(\Delta\theta)$ at the $\tau^{th}$ local step in the $t^{th}$ round is unbiased such that $\mathbb{E}\left[ \|\boldsymbol{g}_{i,t,\tau}(\Delta\theta) - \nabla\mathcal{L}_i(\Delta\theta)\|^2 \right] \leq \sigma^2, \forall i, t, \tau, \Delta\theta \in \mathbb{R}^{d \times r}$.*

**Assumption 4.** *(L-smoothness). The local loss function is L-smoothness such that $\|\nabla\mathcal{L}_i(\Delta\theta_1) - \nabla\mathcal{L}_i(\Delta\theta_2)\| \leq L\|\Delta\theta_1 - \Delta\theta_2\|$ for arbitrary $\Delta\theta_1$ and $\Delta\theta_2 \in \mathbb{E}^{d \times r}$.*

**Assumption 5.** *(Bounded mask discrepancy). The element-wise discrepancy measured by the Hamming distance between any local mask $(dist(\mathbf{m}_t^i, \mathbf{m}_t^j))$, between any local search mask and the optimal local mask of it $(dist(\mathbf{m}_t^i, \mathbf{m}^{i,*}))$, and between any two local optimal masks $(dist(\mathbf{m}^{i,*}, \mathbf{m}^{j,*}))$ are bounded by constants V, Z and U, respectively.*

**Theorem 1.** *(Convergence of PerFIT). Let $N_{ls}$ and $S$ represent the number of local steps and the number of participants in each round, respectively. Given the aforementioned assumptions and static sparsity, assume that the learning rate $\eta \leq \frac{1}{4LN_{ls}}$, the personalized fine-tuning modules $\Delta\theta_{i,t}$ have the following convergence rate:*

$$\frac{1}{Tn} \sum_{t=0}^{T-1} \sum_{i=1}^{n} \mathbb{E}\left[ \|\nabla\mathcal{L}_i(\Delta\theta_{i,t})\|^2 \right] \leq \frac{3\left(f(\Delta\theta_0) - f(\Delta\theta^*)\right)}{\sqrt{T}\eta N_{ls}\kappa} + 3\rho + \epsilon, \tag{6}$$

*where* $\kappa = \frac{1}{2} - 150 N_{ls}^3 \eta^3 L^3 - 15 N_{ls}^2 \eta^2 L^2 - 5 N_{ls} \eta L$, $\rho = (25 N_{ls}^3 \eta^4 L^3 + \frac{5 N_{ls}^2 \eta^3 L^2}{2})(\sigma^2 + 18 N_{ls}\Phi) + \frac{4 N_{ls}^2 \eta^2 L + N_{ls}\eta}{2} Z B^2 + 9 N_{ls}^2 \eta^2 L \Phi + \frac{N_{ls}\eta^2 L \sigma^2}{S}$, $\Phi = (dr/(1-s) - dr)G^2 + B^2(V + Z)$, *and* $\epsilon = 3(dr/(1-s) - dr)C^2 + 3dr B^2 + 3U B^2$.

**Assumptions 1, 2, 3, and 4** follow the commonly used assumptions (10). Existing work (15) has demonstrated that the Hessian of the loss for LLMs shows a small local effective rank, which indicates that the curvature of the loss is constrained along a certain and small number of directions in the parameter space. Since all local clients share the same frozen backbone model which has already learned massive knowledge compared to downstream domain tasks, the curvature differences caused by heterogeneous fine-tuning data are bounded. Note that the NAS metrics defined by Equation 2, 3, and 4 are based on either gradient or Hessian. We assume that the differences in personalized LoRA architectures are bounded as well, which motivates us to make **Assumption 5**. The result in Equation 6 shows that our PerFIT method exhibits the convergence rate of $\mathcal{O}(\frac{1}{T})$. We recall convergence analysis in (10) and display the original formulas as follows based on **Assumptions 1, 2, 3, 4** and **5**.

**Theorem 2.** (***Convergence of the vanilla sparse federated learning (10)***). *Let $N_{ls}$ and $S$ represent the number of local steps and the number of participants in each round, respectively. Given the aforementioned assumptions and static sparsity, assume that the learning rate $\eta \leq \frac{1}{4 L N_{ls}}$, the personalized fine-tuning modules $\Delta\theta_{i,t}$ have the following convergence rate:*

$$\frac{1}{Tn} \sum_{t=0}^{T-1} \sum_{i=1}^{n} \mathbb{E}\left[\|\nabla \mathcal{L}_i(\Delta\theta_{i,t})\|^2\right] \leq \frac{3(f(\Delta\theta_0) - f(\Delta\theta^*))}{\sqrt{T}\eta N_{ls}\kappa} + 3\rho + \epsilon, \tag{7}$$

*where* $\kappa = \frac{1}{2} - 150 N_{ls}^3 \eta^3 L^3 - 15 N_{ls}^2 \eta^2 L^2 - 5 N_{ls} \eta L$, $\rho = (25 N_{ls}^3 \eta^4 L^3 + \frac{5 N_{ls}^2 \eta^3 L^2}{2})(\sigma^2 + 18 N_{ls}\Phi) + \frac{4 N_{ls}^2 \eta^2 L + N_{ls}\eta}{2n} \sum_n (dist(\mathbf{m}_t^i, \mathbf{m}^{i,*})) B^2 + 9 N_{ls}^2 \eta^2 L \Phi + \frac{N_{ls}\eta^2 L \sigma^2}{S}$, $\Phi_t = \frac{1}{n} \sum_i ((dr/(1-s) - dr)G^2 + \frac{1}{n}\sum_j B^2 (dist(\mathbf{m}_t^i, \mathbf{m}_t^j) + dist(\mathbf{m}_t^j, \mathbf{m}^{j,*})))$, *and* $\epsilon = 3(dr/(1-s) - dr)G^2 + 3dr B^2 + \frac{3}{n^2} \sum_i \sum_j dist(\mathbf{m}^{i,*}, \mathbf{m}^{j,*}) B^2$.

By ignoring the $\frac{1}{T}$ and $\frac{1}{T^{2/3}}$ terms and substituting the dynamic mask similarities in the original with our static mask similarities defined by **Assumption 5**, we can derive the convergence rate of our PerFIT method as $\mathcal{O}(\frac{1}{\sqrt{T}})$.

# 5 EXPERIMENTS

## 5.1 EXPERIMENTAL SETTINGS

**Dataset.** We conducted our experiments on four datasets: Databricks-dolly-15k (5), MedAlpaca (8), CodeAlpaca (2), and MathInstruct (31). Databricks-dolly-15k is a general instruction-following dataset, including creative writing, brainstorming, classification, closed QA, generation, information extraction, open QA, and summarization. MedAlpaca, Code-Alpaca, and MathInstruct are domain-specific instruction-following dataset. We performed two types of splitting methods to emulate the heterogeneous data distributed to local clients. The first is the *pathological* non-IID setup where each client is randomly assigned the same number of data points. For Databricks-dolly-15k, we randomly assigned 2 classes among 8 total classes to each client. For other domain-specific datasets, we randomly assigned 200 data to each client. The second non-IID setup follows the *Dichilet* distribution, which is parameterized by a coefficient $\beta$, denoted as Dir($\beta$). $\beta$ determines the degree of data heterogeneity. The smaller the $\beta$ is, the more heterogeneous the data distributions will be. We set the $\beta$ as 0.5 throughout the experiments. Since the Databricks-dolly-15k is the only dataset that has labels, we only apply the *Dichilet* method on it.

**Models and Baselines.** To showcase the effectiveness of our method on various LLMs, we utilized three open-source large language models: Alpaca-7B (22), Vicuna-7B-v1.5 (4) and LLaMA-2-7B (23). The first two LLMs have been fine-tuned based on the LLaMA-1-7B (24) to enhance their abilities to understand and respond to human inputs effectively while the LLaMA-2-7b model has not been. We used the official tokenizers that correspond to the model weights. We developed our method based on two federated instruction tuning frameworks: Federatedgpt (32) and OpenFedLLM(29). Note that the mentioned two methods focus on obtaining a global fine-tuned

model. They also show that federated instruction tuning is better than only using local data to fine-tune. Since our method is the first solution to focus on the personalized federated instruction tuning problem, we only compare it against the global model.

**Configurations.** For all experiments, we set the number of total clients as 100. The backbones of the LLMs are frozen during pruning and local fine-tuning to save the memory cost. We add LoRA to three attention modules for every layer, i.e., *Query*, *Key*, and *Value* matrices. For homogeneous resource baselines, we set the basic rank $r$ of LoRA as 8. The coefficient $alpha$ remains the same value of $r$ for all experiments. Note that the comparisons are under the prerequisite that all methods have the same number of trainable parameters. The sparsity levels for our method are designated as 0.66 and 0.5, corresponding to the original ranks of 12 and 16. For heterogeneous scenarios, we categorize the capability of clients into three levels: i) Large; ii) Medium; and iii) Small. Each category has $1/3$ of the total number of clients.

## 5.2 PERFORMANCE EVALUATION

Table 1: Perplexity comparison on domain-specific datasets.

| Model | Sparsity | Dataset | | | | | |
| | | MedAlpaca | | CodeAlpaca | | MathInstruct | |
| | | FIT | PerFIT | FIT | PerFIT | FIT | PerFIT |
|---|---|---|---|---|---|---|---|
| Alpaca-7B | 0.33 | 2.18 | 2.13(-0.05) | 1.86 | 1.84(-0.02) | 2.79 | 2.51(-0.28) |
| | 0.50 | | **2.10(-0.08)** | | **1.83(-0.03)** | | **2.48(-0.31)** |
| Vicuna-7B | 0.33 | **1.92** | 1.93(+0.01) | 1.78 | 1.77(-0.01) | 2.40 | 2.27(-0.13) |
| | 0.50 | | **1.92(-0.00)** | | **1.76(-0.02)** | | **2.22(-0.18)** |
| LLaMA-2-7B | 0.33 | **1.88** | 1.89(+0.01) | 1.74 | 1.73(-0.01) | 2.27 | 2.19(-0.08) |
| | 0.50 | | **1.88(-0.00)** | | **1.73(-0.01)** | | **2.18(-0.09)** |

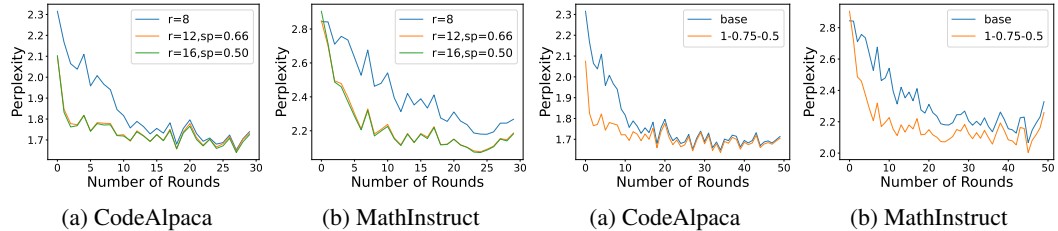

| (a) CodeAlpaca | (b) MathInstruct | (a) CodeAlpaca | (b) MathInstruct |
|---|---|---|---|

Figure 2: Perplexity for homogeneous resources.    Figure 3: Perplexity for heterogeneous resources.

**Performance on Homogeneous Resources.** Table 1 presents the results of the perplexity comparison under homogeneous resource scenarios on domain-specific datasets. Our method achieves the largest, and the second largest perplexity reduction on the MathInstruct and CodeAlpaca, respectively. The largest perplexity reduction is 11.1%, which is obtained based on the Alpaca-7B model. On the MedAlpaca dataset, however, we only observe nearly zero perplexity decreases. We further present the learning curves of LLaMA-2-7B in Figure 2. We can observe that our PerFIT method converges to the same perplexity level on the MedAlpaca dataset. On the CodeAlpaca and MathInstruct datasets, our method invariably exhibits fast convergences and smaller perplexity values.

Table 2 presents the results of the perplexity comparison on the Databricks-dolly-15k dataset. The perplexity achieved by implementing our method consistently outperforms the vanilla FIT method. For the pathological none-IID setting, PerFIT on the Alpaca model with the original rank 12 and 16 outperforms FIT by 23% and 9%, respectively. Under the same non-IID setting, the perplexity results of the Vicuna model with rank 12 and 16 decrease by

Table 2: Perplexity comparison on Databricks dataset.

| Dis. | Model | Sparsity | Methodology | |
| | | | FIT | PerFIT |
|---|---|---|---|---|
| Path. | Alpaca | 0.33 | 5.15 | **3.93(-1.22)** |
| | | 0.50 | | 4.66(-0.49) |
| | Vicuna | 0.33 | 4.22 | **4.09(-0.13)** |
| | | 0.50 | | **4.09(-0.13)** |
| Dir. | Alpaca | 0.33 | 5.28 | **4.13(-1.15)** |
| | | 0.50 | | 4.71(-0.57) |
| | Vicuna | 0.33 | 3.85 | 3.81(-0.04) |
| | | 0.50 | | **3.78(-0.07)** |

3%. For the Dirichlet (0.5) non-IID scenario, our method improves the Alpaca model by 21% and 10%, respectively. For the Vicuna model under the Dirichlet setting, our PerFIT method reduces the perplexity by 1% for both rank 12 and 16 settings, respectively.

**Performance with Higher Sparsity Values.** To explore the performance of our method at higher sparsity values, we extended our experiments with a rank of $24$, which corresponds to a sparsity value of $0.33$, on the Databricks-dolly-15k dataset. Figure 4 shows the perplexity curves. We consistently observe fast convergence and lower losses with all rank settings. The curve with a rank of $12$ converges to the smallest value of perplexity on two different non-IID settings. For the Vicuna model, we find that our PerFIT method invariably enjoys a fast convergence speed at the early stage on all rank settings. The curve with a sparsity value of $0.66$ exhibits the best overall performances considering both convergence speed and perplexity value.

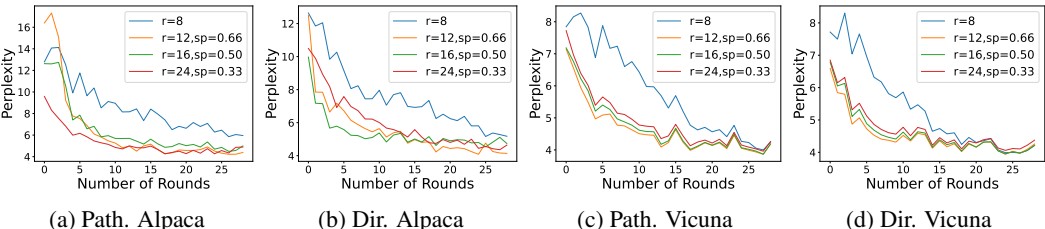

(a) Path. Alpaca      (b) Dir. Alpaca      (c) Path. Vicuna      (d) Dir. Vicuna

Figure 4: Perplexity for homogeneous resources.

**Performance on Heterogeneous Resources.** Table 3 shows the perplexity results of heterogeneous resources on the Databricks-dolly-15k dataset. By utilizing the proposed architecture search and personalized aggregation methods, we can observe that the PerFIT method facilitates local fine-tuning within heterogeneous resource scenarios. It is worth noting that the Vicuna still behaves better than the Alpaca model on resource heterogeneity scenarios. Under the pathological non-IID setting, our method shows a $12\%$ decrease in perplexity

Table 3: Perplexity comparison on heterogeneous resources.

| Dis. | Model | Methodology | |
|---|---|---|---|
| | | FIT | PerFIT |
| Path. | Alpaca | 4.48 | **3.93(-0.55)** |
| | Vicuna | 3.78 | **3.63(-0.15)** |
| Dir. | Alpaca | 4.17 | **4.05(-0.12)** |
| | Vicuna | 3.70 | **3.52(-0.18)** |

compared to the FIT on the Alpaca model. For the Vicuna model, we can observe a $3\%$ reduction in perplexity. With Dirichlet configuration, our method improves the perplexity by $2\%$ and $4\%$ on the Alpaca and Vicuna models, respectively. Figure 3 and 5 display the learning curves on the domain-specific and Databricks-dolly-15k datasets, respectively. "base" represents the results obtained with rank $8$. "$1 - 0.75 - 0.5$" represents the performance of our PerFIT method. We can observe that our method significantly improves the performance of personalization on all datasets, proving that our method can not only allow collaborative fine-tuning for resource heterogeneous clients but also boost the overall personalization performance.

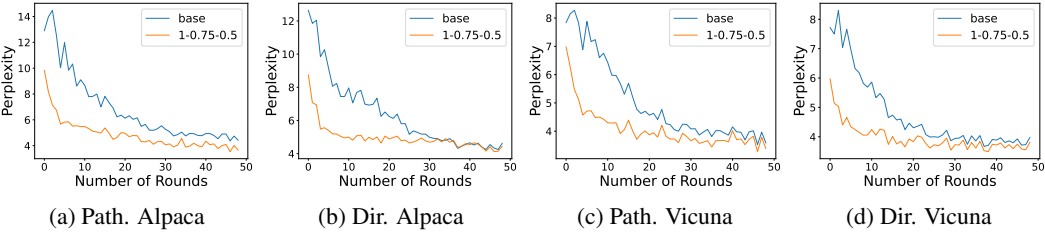

(a) Path. Alpaca      (b) Dir. Alpaca      (c) Path. Vicuna      (d) Dir. Vicuna

Figure 5: Perplexity for heterogeneous resources on Databricks-dolly-15k.

**Different Numbers of Participants.** To demonstrate the scalability of our method across various numbers of participants in each round, we conducted extensive experiments by randomly selecting $5\%$ and $20\%$ clients in each round under the Dirichlet non-IID settings. For the Alpaca model, we can observe that our method Similar to the results shown in Figure 4, we observe that our method implemented on the Alpaca model displays more notable performance improvements. For the Vicuna model, we find that our method converges to the same value as that of FIT but with a remarkable increase in the speed of convergence.

**Mask Similarity Analyses.** Figure 7 shows the pair-wise mask similarity between the first LoRA modules of $10$ clients randomly selected. The rank is set to $16$ and the sparsity is set to $0.50$. The labels of the x and y-axes represent the index of the client. The similarity is measured by the Hamming distance. We can observe that clients with heterogeneous data own personalized

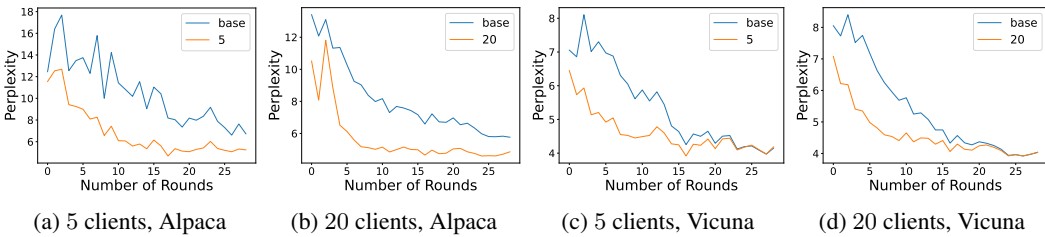

| (a) 5 clients, Alpaca | (b) 20 clients, Alpaca | (c) 5 clients, Vicuna | (d) 20 clients, Vicuna |

Figure 6: Perplexity for different # of local participants.

masks. Furthermore, the degree of any pair-wise similarity is close across clients, which supports and reinforces our assumption of bounded mask discrepancy.

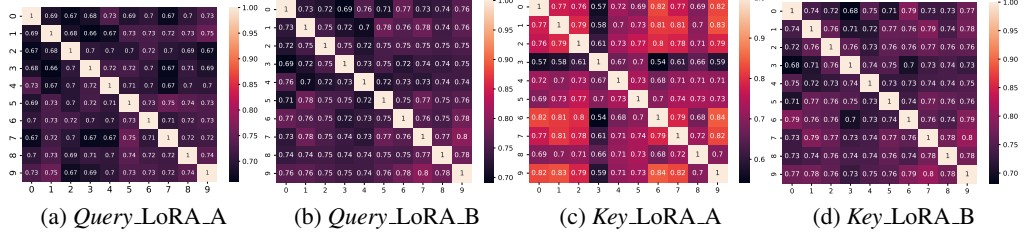

| (a) *Query*_LoRA_A | (b) *Query*_LoRA_B | (c) *Key*_LoRA_A | (d) *Key*_LoRA_B |

Figure 7: Comparison of mask similarities.

**Impact of Important Score Metric.** To evaluate the impact of using different metrics for computing the important scores, we further conducted experiments based on the Vicuna model under pathological non-IID settings. The rank is set to 16 with a sparsity of 50%. The comparisons are shown in Figure 8. The "first", "second", and "mix" curves denote the results obtained based on Equation 2, 3, and 4, respectively. We can observe that all metrics exhibit extremely similar training dynamics. Since the second-order information requires extra computation overhead, we recommend using the first-order metric in practice.

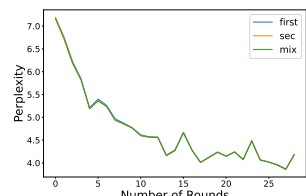

Figure 8: Comparison for pruning metrics.

**Impact of Initialization.** To evaluate the impact of different initialization schemes, we conducted experiments using the model Alpaca-7B and the dataset Databricks-dolly-15k with the pathological non-IID distribution. We initialized the Alpaca-7B model using either the uniform or the normal distribution. We conducted FL based on standard FIT (with a rank of 8) and our PerFIT (with a rank of 12), respectively. For FIT, the final perplexity of the case with uniform initialization is 0.35 smaller than that of the case with normal initialization, where the two cases have the same convergence rates. For PerFIT, the case with uniform initialization achieves a perplexity of 2.96, while the case with normal initialization achieves a perplexity of 3.93. We can observe that our PerFIT method consistently outperforms the FIT method under different initialization methods. Note that the two PerFIT cases have better convergence rates than their FIT counterparts. Therefore, our proposed method consistently outperforms FIT under different initialization strategies.

## 6 CONCLUSION

While federated instruction tuning has demonstrated the ability to improve global model performance without revealing private instruction-following data, this approach fails to address the issues of personalized data and varying client resources. In this paper, we introduced a novel personalized federated instruction tuning named PerFIT. By enabling local clients to search for personalized fine-tuning architectures in an expanded LoRA space, we effectively mitigate the difficulties posed by heterogeneous data and resource distributions We analyzed the convergence property of our method, showing that our method tailored for LLMs exhibits a similar convergence rate to the sparse federated training method. Comprehensive experimental results on representative LLMs under two non-IID scenarios demonstrated the effectiveness of our proposed method.

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

# A DATASETS DESCRIPTION

For datasets except Databricks-dolly-15k (5), we randomly sample 20K to form our fine-tuning data. We display the format of the datasets and corresponding links here:

1. Databricks-dolly-15k (5) is a corpus of more than 15K records generated by thousands of Databricks employees to enable large language models to exhibit the magical interactivity of ChatGPT.

2. MedAlpaca (8) has 34K records and addresses subjects such as anatomy, physiology, pathology, pharmacology, and more.

3. CodeAlpaca (2) has 20K records and aims to build and share an instruction-following LLaMA model for code generation.

4. MathInstruct has more than 225K data points and is compiled from 13 math rationale datasets.

# B PROMPT TEMPLE AND DATA EXAMPLE

We follow the prompt used in Alpaca (22) and show it in Table 4. We show one example from the MedAlpaca dataset in Table 5.

Table 4: Prompt template.

| |
|---|
| Below is an instruction that describes a task. Write a response that appropriately completes the request. 
 ### Instruction: {Instruction} 
 ### Response: |

Table 5: Data example.

| |
|---|
| Below is an instruction that describes a task. Write a response that appropriately completes the request. 
 ### Instruction: What is the endocrine tumor that is identified by small nests of spindle-like chromaffin cells "Zellballen" and granular cytoplasm? 
 ### Response: Pheochromocytoma is the endocrine tumor that is characterized by granular cytoplasm and small nests of spindle-like chromaffin cells "Zellballen". Pheochromocytoma is a rare neuroendocrine tumor that arises from chromaffin cells in the adrenal medulla. These tumors produce excess amounts of catecholamines, which can lead to episodes of hypertension, headaches, palpitations, and sweating. Histologically, pheochromocytomas are characterized by small nests of spindle-like chromaffin cells called "Zellballen", which are surrounded by fibrous stroma. The cells have granular cytoplasm and round to oval nuclei with prominent nucleoli. The diagnosis of pheochromocytoma typically involves blood and urine tests to measure levels of catecholamines and their metabolites, as well as imaging studies such as CT or MRI scans. Treatment may involve surgery to remove the tumor, as well as medications to control blood pressure and manage symptoms. It is important to note that the histological appearance of pheochromocytoma can vary, and some tumors may not have the characteristic "Zellballen" pattern. Therefore, the diagnosis of pheochromocytoma should not be based solely on histological appearance, and clinical and biochemical data should also be taken into consideration. |

# C HYPERPARAMETERS

All experiments were run on one NVIDIA RTX 4090 GPU. We set the rank for clients with the smallest capability as 8. Therefore, the rank for Medium and Large is set to 12 and 16, respectively. We set 5 for the number of local pruning epochs. In each round of local fine-tuning, we randomly select 10% of clients. For all experiments, the local batch size is set to 64. To facilitate training with batched data on a single GPU, we utilize the gradient accumulation with a mini-batch size of 8. The total training rounds are 30 and 50 for homogeneous and heterogeneous scenarios, respectively. The

local training epoch is 1. We split $80\%$ of local data into training and use the rest to evaluate the performance of personalization. The average perplexity in each round is reported. Please refer to Appendix C for details.

We followed the configuration in OpenFedLLM (29) and used the Adam optimizer with a cosine learning rate schedule based on the index of the training round. We set the initial learning rate, and the minimum learning rate to $5e-5$ and $1e-6$, respectively. The $\beta_1$, $\beta_2$ and $\epsilon$ are set to 0.9, 0.99, and $1e-8$, respectively.

