# OpenReview forum: "PerFIT: Personalized Federated Instruction Tuning via Neural Architecture Search"
_ICLR.cc/2025/Conference — Submitted to ICLR 2025_

### Official Review · Reviewer_yZsM · 2024-10-21

**Soundness:** 4
**Presentation:** 3
**Contribution:** 3
**Rating:** 8
**Confidence:** 3

**Summary:**

The paper presents PerFIT, a framework that personalizes federated tuning of LLMs to tailor model architectures to clients' needs. It introduces a parameter-wise aggregation strategy to balance personalization and collaboration, ensuring effective model updates despite data and resource heterogeneity. PerFIT demonstrates improvements in perplexity and scalability across datasets, outperforming standard federated learning baselines. This contribution addresses key challenges in decentralized learning, such as model adaptation and communication efficiency, offering a scalable solution for diverse federated environments.

**Strengths:**

* The use of NAS helps design sparse and client-specific architectures. It ensures that even with limited client resources, models remain effective by focusing on personalized tuning. Models can achieve better convergence on client datasets without forcing a universal model structure.

* Instead of aggregating entire models, PerFIT uses parameter-wise strategies. This enables more efficient collaboration between clients while maintaining the benefits of local personalization, reducing the risk of model degradation across heterogeneous clients.

**Weaknesses:**

* The paper does not explore the scalability of PerFIT with models larger than 7B parameters or client populations beyond the tested settings. More experiments with larger-scale models and diverse client distributions could strengthen the claims, especially regarding the framework’s computational and time complexities.

* A deeper analysis of communication costs and performance consistency across varying client heterogeneity could offer further insights into PerFIT’s applicability and robustness under real-world conditions.

**Questions:**

* How does PerFIT perform when client datasets are drastically imbalanced or highly diverse? Are there scenarios where parameter-wise aggregation fails to converge or leads to suboptimal results?

* How does PerFIT perform with larger models beyond 7B size, would the computational burden of PerFIT increase linearly? Are there any planned optimizations to prevent such overhead?

---

> ### Author Response · Authors · 2024-11-22
>
> **Response to Weakness 1:**
>
> *Experiments on Llama-2-13B*:
>
> Thank you for your suggestions. We have conducted experiments based on Llama-2-13B with the CodeAlpaca dataset. The training rounds are set to 10 and the rest of the hyperparameters remain the same as we used in our paper. The perplexity values for FIT and our PerFIT are 1.782 and 1.780, showing that our PerFIT method consistently outperforms the baseline method on larger LLMs.
>
> *Experiments on diverse client distributions*:
>
> To form diverse client distributions, we followed the dataset used in FlexLoRA [1] and conducted more experiments. The dataset consists of over 1600 distinct natural language tasks that come from 76 NLP task types and is split based on its meta-info of the belonging NLP tasks, such that each client holds a unique task to mirror a task heterogeneous environment.
> We conduct experiments based on the model and Rouge-L score used in [1]. Note that the Rouge-L score is positively correlated with the human evaluation result, meaning that it is a valid indicator for evaluating the performance of LLMs [1]. We keep the base rank of LoRA as 8. The rank and sparsity rate for our PerFIT is set to 12 and 33%, respectively. The rest hyperparameters follow the configuration defined in our PerFIT paper.
> The Rouge-L scores for standalone, FIT and our PerFIT are 34.06, 55.15 and 57.05, respectively. We can observe that our method consistently outperforms baseline methods for personalization.
>
>
>
> **Response to Weakness 2:**
>
> Thank you for your suggestions. The communication costs of our methods consist of two parts: i) parameter transmission and ii) mask transmission.
> The costs of parameter transmission are the same as the baseline method FIT. The binary masks are stored in the bitmap data structure to save the communication overhead. Furthermore, the masks only need to be transmitted once after the local NAS as shown in Figure 1. Therefore, the communication burden on mask transmission is negligible compared to the parameter transmission and the overall communication costs remain the same as the baseline method FIT.
>
> **Response to Question 1:**
>
> We have conducted new experiments on more diverse client datasets. Please refer to **Response1 for Weakness1**, *Experiments on diverse client distributions* for the experimental results and discussions.
>
> Our parameter-wise aggregation is robust to heterogeneity scenarios from both experimental and theoretical perspectives. The proposed personalized aggregation strategy is designed to handle varying degrees of heterogeneity. We provide theoretical analyses (lines 313-337) and experimental verifications (lines 483-496) that demonstrate the bounded nature of heterogeneity in the context of LLM-based fine-tuning.
>
>
> **Response to Question 2:**
>
> Thank you for your suggestions. We have conducted experiments on the Llama-2-13B model and verify the effectiveness of our method. Please refer to **Response1 for Weakness1**, *Experiments on Llama-2-13B* for the experimental results and discussions.
>
> Since our method performs the NAS only on LoRA layers, the computation burden of NAS relies on the number of attention layers and increases linearly as we utilize a larger model like Llama-2-7-13B. We plan to integrate the sparse matrix multiplication acceleration method [2] to relieve the computational burden.
>
>
> [1] Federated Fine-tuning of Large Language Models under Heterogeneous Tasks and Client Resources. NeurIPS 2024.
>
> [2] SparTA: Deep-Learning Model Sparsity via Tensor-with-Sparsity-Attribute. OSDI 2022.

---

> > ### Comment · Reviewer_yZsM · 2024-11-25
> >
> > Thank you for your detailed response. This reply addresses my concerns and provides evidence to support the claims.

---

> > > ### Author Response · Authors · 2024-11-30
> > >
> > > Thank you very much for your feedback.
> > > We genuinely value your insightful feedback and the acknowledgment of our efforts.

---

### Official Review · Reviewer_dUBm · 2024-10-28

**Soundness:** 2
**Presentation:** 3
**Contribution:** 2
**Rating:** 3
**Confidence:** 4

**Summary:**

This work primarily addresses the challenges of data distribution heterogeneity and client-side computational resource heterogeneity in federated instruction tuning. To tackle these issues, it proposes a neural architecture search-based approach to achieve personalized model parameters and structures. Data heterogeneity and resource heterogeneity are classic challenges in FL community, with substantial prior research dedicated to them. Revisiting these problems in the era of LLMs is meaningful. However, the work’s technical and theoretical contributions are not prominent, and there is also a lack of empirical analysis to support the stated research challenges. Additionally, the experimental evaluation has areas for improvement. Overall, I believe this paper is not suitable for acceptance, especially at a top-tier conference like ICLR.

**Strengths:**

1. This works is targeted to valuable issues in federated instruction tuning.
2. Figure 1 is well presented to make a clear introduction of the proposed approach.

**Weaknesses:**

1. The authors claimed that this work is motivated by "intrinsic connection between data heterogeneity and architecture heterogeneity", but did not present clear evidence for the intrinsic connection. If no intrinsic connection exists, this work actually solves two separated problems. Given the existence of corresponding work to address these issues, the novelty of this proposed work is diminished ([1] for resource heterogeneity, and [2] for benchmarking personalized approaches for data heterogeneity in federated LLM tuning).
2. I am curious whether this masking approach can truly address the issue of resource heterogeneity. Masking does not seem to reduce the computational load since current libraries lack satisfactory support for masked models in terms of computational efficiency. If the authors emphasize their method’s contribution to resource heterogeneity, supporting experimental results are needed.
3. A key goal of the proposed method is to address resource heterogeneity, claiming that previous FL fine-tuning work based on PEFT methods such as LoRA mainly involved homogeneous models. However, from Figure 1, it appears that only the LoRA components were subjected to NAS. Given that the parameters in LoRA only occupy a small portion of the entire LLM, I am curious about the extent to which this method actually contributes for solving heterogeneous computational resources on the client side.
4. In line 703, the authors claim that "The average perplexity in each round is reported. Please refer to Appendix C for details". However, I couldn’t find the corresponding information. Also, why was I directed from Appendix C to look for content in Appendix C?
5. Randomly assigning 200 data samples to each client represents a highly unrealistic scenario, where the data distribution is IID, and even the quantity of data is also IID (line 367). Experiments conducted under this scenario constitute the majority of the experimental evaluation, which somewhat undermines the persuasiveness of the method's effectiveness.
6. Since this work focuses on personalized FL, comparing only with the FIT method is insufficient. On one hand, more advanced personalized FL fine-tuning methods should be included for comparison, such as [3]. On the other hand, it is recommended to fine-tune the LLM obtained by FIT to adapt it as a personalized federated approach.
7. The authors demonstrate the convergence of their method. This type of analysis has already been extensively conducted in traditional FL studies. Considering that the theoretical modeling in this manuscript does not differ from traditional FL or masked-based FL, it is debatable whether dedicating substantial space to this well-established theoretical analysis is truly necessary. Moreover, whether LLMs genuinely satisfy the L-smoothness assumption remains a contentious issue, which makes the theoretical contribution of this paper less significant.


[1] Federated Fine-tuning of Large Language Models under Heterogeneous Tasks and Client Resources. NeurIPS 2024.

[2] Federatedscope-LLM: A comprehensive package for fine-tuning large language models in federated learning. KDD 2024.

[3] FDLoRA: Personalized Federated Learning of Large Language Model via Dual LoRA Tuning. arXiv24.

**Questions:**

Please refer to Weaknesses.

---

> ### Author Response · Authors · 2024-11-22
>
> **Response to Weakness 1:**
>
> The improvements shown in experiments provide evidence for the intrinsic connection. Our NAS process is conducted for every client based on the specific local dataset. The overlap between local masks represents the parameters for the global training objective. The dissimilarities between masks demonstrate the connection between data heterogeneity and architecture heterogeneity. Therefore, our method provides a novel approach that can simultaneously address the data and resource heterogeneity challenges.
>
> **Response to Weakness 2:**
>
> Thank you for your suggestions. Since existing work [1] has already provided unstructured sparsity acceleration for DNN training under a wide range of sparsity, considering that the LoRA structures are MLP layers, the integration of [1] and our unstructured sparsity method can truly address the issue of resource heterogeneity.
>
>
> **Response to Weakness 3:**
>
> Our method does contribute to solving heterogeneous computational resources on the client side.
> Although the LoRA structure accounts for about 1% of the parameters of the backbone of LLM, the backbone is frozen with advanced quantization methods while the LoRA remains high precision floating points. Moreover, the application of LoRA in traditional FL methods suffers from the “bucket effect”, meaning that clients with more fine-tuning resources have to compromise to the smallest viable LoRA rank [2].
>
>
> **Response to Weakness 4:**
>
> Sorry for the confusion. "The average perplexity in each round is reported” means that the results in all experiments are the averaged local perplexity. Please disregard the phrase "Please refer to Appendix C for details," as it was mistakenly included. We will revise this part in the final version.
>
>
> **Response to Weakness 5:**
>
> Thanks for your suggestions.
> We have conducted new experiments based on the dataset, model and Rouge-L score used in [2] to show the advantages of our method.
> According to [2], the dataset consists of over 1600 distinct natural language tasks that come from 76 NLP task types and is split based on its meta-info of the belonging NLP tasks.
> Note that the Rouge-L score is positively correlated with the human evaluation result, meaning that it is a valid indicator for evaluating the performance of LLMs [2]. We keep the base rank of LoRA as 8. The rank and sparsity rate for our PerFIT is set to 12 and 33%, respectively. The rest hyperparameters follow the configuration defined in our PerFIT paper. The Rouge-L scores for FIT and our PerFIT are 55.15 and 57.05, showing that our method achieves the best performance by increasing the Rouge-L score by 1.9.
>
>
> **Response to Weakness 6:**
>
> To fairly compare our method with existing work and show the effectiveness of our method, we have reproduced the method in [2] and compared the performance of personalized based on the dataset, model and evaluation metric used in [2]. We randomly assign rank 8, 12, and 16 to 100 clients. In each fine-tuning round, we randomly select 10 clients for local fine-tuning. The number of fine-tuning rounds is set to 50 and the rest hyperparameters follow the configurations used in our work. The averaged Rouge-L score for [2] and our method is 40.17 and 58.28, respectively, showing that our method outperforms the method in [2].
>
> According to the ICLR 2025 reviewer instruction: “Authors are encouraged to cite and discuss all relevant papers, but they may be excused for not knowing about papers not published in peer-reviewed conference proceedings or journals, which includes papers exclusively available on arXiv.”, we thoroughly read work [3]. Unfortunately, [3] was only published on arXiv and has not released code and been peer-reviewed, Unfortunately, [3] was only published on arXiv and has not released any code or undergone peer review, so we are unable to reproduce the results.
> Note that the experimental results obtained by comparing with [2] have already shown that our method outperforms the method in [2].  We believe the comparison results are enough to exhibit the superiority of our method.
>
>
> **Response to Weakness 7:**
>
> We provided convergence analyses to demonstrate that our method achieves the same convergence rate as previous methods [4], thereby showing that there is a solid theoretical foundation for our approach. Moreover, we delved into the analyses of the loss landscape of pre-trained LLM in lines 329-337 and proposed our assumptions on bounded mask discrepancy. The mask similarity analysis in lines 483-496 further supports our assumptions.
>
> [1] SparTA: Deep-Learning Model Sparsity via Tensor-with-Sparsity-Attribute. OSDI 2022.
>
> [2]Federated Fine-tuning of Large Language Models under Heterogeneous Tasks and Client Resources. NeurIPS 2024
>
> [3] FDLoRA: Personalized Federated Learning of Large Language Model via Dual LoRA Tuning. arXiv24.
>
> [4] Achieving personalized federated learning with sparse local models. arXiv22.

---

> > ### Comment · Reviewer_dUBm · 2024-11-27
> >
> > Thanks for your response. Based on your response, my concerns have still not been addressed, especially regarding the significance of this work as mentioned in Weakness 2 and Weakness 3. Therefore, I decide to keep my original score.

---

> > > ### Author Response · Authors · 2024-11-30
> > >
> > > Thank you for your response.  We want to further address your concerns from the following two aspects:
> > >
> > > **(1) The effectiveness of using unstructured sparse masks to reduce computation load.**
> > >
> > > **(2) The significance of using heterogeneous ranks of LoRA to solve heterogeneous computational resources.**
> > >
> > >
> > > For **(1)**, we have already shown that existing work [1] has provided an unstructured sparsity acceleration library for MLP layers under a wide range of sparsity. Since the LoRA structures are MLP layers, integrating [1] and our unstructured sparsity method can truly reduce the computation load.
> > >
> > > For **(2)**, we have demonstrated that exploring heterogeneous ranks of LoRA across local clients is meaningful. This is because the "bucket effect" [2-3] is prevalent in traditional LoRA-based federated learning methods, where clients with more fine-tuning resources have to compromise to accommodate the smallest viable LoRA rank. To address the challenge of heterogeneous resources, our method utilizes sparse matrices, while [2] and [3] focus on matrix decomposition.
> > >
> > >
> > > [1] Improving LoRA in Privacy-preserving Federated Learning. ICLR 2024.
> > >
> > > [2] Heterogeneous LoRA for Federated Fine-tuning of On-Device Foundation Models. EMNLP 2024.
> > >
> > > [3] Federated Fine-tuning of Large Language Models under Heterogeneous Tasks and Client Resources. NeurIPS 2024.

---

### Official Review · Reviewer_9rUg · 2024-10-29

**Soundness:** 3
**Presentation:** 3
**Contribution:** 2
**Rating:** 3
**Confidence:** 4

**Summary:**

This work proposes a NAS-based method to solve the data and resource heterogeneity faced by previous federated instruction methods which adopt a unified global model. To evaluate the effectiveness of the proposed approach, experiments are performed on four instruction datasets, with a native federated instruction tuning method as the baseline. These challenges are reasonable; however, the innovation of the method is limited, and the brought improvements are also minor.

**Strengths:**

1. This work is well-organized and well-presented, making it easy to follow.
2. Code is available. Although no documentation is provided to explain how to run the experiments, having the code is certainly better than having none at all.

**Weaknesses:**

1. Marginal improvements on performance. From Table 1, PerFIT exhibits a real small improvement on perplexity compared to FIT. Considering the wide range of values that perplexity can take, I am doubtful whether this slight improvement obtained by PerFIT actually contributes to enhancing the LLMs' performance. The authors could provide examples or analyses demonstrating how these small perplexity improvements translate to practical enhancements in LLM performance.
2. Lacks of novel contribution. Although the problem addressed in this work is meaningful, the proposed method does not show a significant distinction from traditional methods, i.e., it seems to merely change the application context from traditional FL for small models to LLMs fine-tuned with LoRA.
3. The first paragraph in the second section is entitled with "Federated Instruction Tuning of Large Language Models". However, the majority of this paragraph discusses matters unrelated to FL, making the inclusion of this paragraph perplexing.
4. This paper overclaims its contributions to the issue of resource heterogeneity. The method is based on LoRA, which typically accounts for only about 1% of the parameters of a full LLM. In this context, the gains from reducing the number of parameters through masking are minimal, regardless of whether it concerns computation, communication, or memory overhead. The authors should clearly quantify how much the resource heterogeneity could be enabled by the proposed approach.

**Questions:**

1. What is the essential difference between the proposed method and existing NAS-based FL methods?
2. Does adding a mask to LoRA adapters affect the consistency between the training objective and the FL objective?
3. From the benchmarking results in [1], the heterogeneity of data distribution seems to affects the fine-tuned results slightly. Does this affect the significance of addressing data distribution heterogeneity in personalized federated instruction tuning to some extent?

[1] FederatedScope-LLM: A Comprehensive Package for Fine-tuning Large Language Models in Federated Learning.

---

> ### Author Response · Authors · 2024-11-22
>
> **Response to Weakness 1:**
>
> Thank you for your suggestions. Since the small perplexity value decrease in our method might not contribute to improving the LLM’s performance, we further conduct experiments based on the dataset, model and Rouge-L score used in [3] to show the advantages of our method. Note that the Rouge-L score is positively correlated with the human evaluation result, meaning that it is a valid indicator for evaluating the performance of LLMs [3]. We keep the base rank of LoRA as 8. The rank and sparsity rate for our PerFIT is set to 12 and 33%, respectively. The rest hyperparameters follow the configuration defined in our PerFIT paper. The Rouge-L scores for FIT and our PerFIT are 55.15 and 57.05, showing that our method achieves the best performance by increasing the Rouge-L score by 1.9.
>
> **Response to Weakness 2:**
>
> Our method is not a direct transfer from the traditional FL method to LLMs fine-tuned with LoRA.  Instead, we proposed a novel method that first expands the rank of LoRA and then searches for personalized parameters for each client by using NAS. The "expanding-searching" scheme greatly differs from the traditional FL sparse training methods.
>
> **Response to Weakness 3:**
>
> Sorry for the confusion. We will revise this paragraph and discuss more related work about the instruction tuning of LLMs in the context of federated learning.
>
>
> **Response to Weakness 4:**
>
> Although the LoRA structure accounts for about 1% of the parameters of the backbone of LLM, the backbone is frozen with advanced quantization methods while the LoRA remains high precision floating points. Moreover, the application of LoRA in traditional FL methods suffers from the “bucket effect”, meaning that clients with more fine-tuning resources have to compromise to the smallest viable LoRA rank according to [3].
>
> Our method does not aim to reduce the number of parameters through masking. Instead, we strictly follow the objective of NAS, which is searching for the best architecture given a specific resource. We establish the search space by expanding the rank of LoRA (Lines 152-153). Then we search for a personalized sparse pattern of parameters for each client through pruning. With our NAS method and aggregation strategy, the clients that own different training fine-tuning resources can successfully conduct parameter sharing.
>
> In our experiments in lines 330-345, we simulated the resource heterogeneity by randomly allocating rank 8, 12, and 16 to 100 clients. The experimental results showed that our method can not only allow collaborative fine-tuning for resource heterogeneous clients but also boost the overall personalization performance.
>
>
> **Response to Question 1:**
>
> Existing NAS-based FL methods (e.g., [2]) focus on searching the architecture based on the module candidate pools. The searched network will be trained from scratch to achieve the best performance. However, the NAS on modules in the era of LLMs fine-tuning is too expensive since the cost of pre-training is unaffordable. Therefore, we focus on LoRA structures and propose a novel method that first expands the rank of LoRA and then searches for personalized parameters for each client by using the NAS, aiming to find the sparse pattern of trainable parameters in LoRA structures.
>
> **Response to Question 2:**
>
> The consistency between the two objectives is not affected. Our method searches the specific sparse pattern for every client. With our theoretical assumptions 4 and the corresponding mask similarity analyses, we can find that the overlap between masks represents the FL training objective and those personalized parameters represent the local training objective.
>
> **Response to Question 3:**
>
> The results in [1] do not diminish the significance of addressing data heterogeneity in personalized federated instruction tuning. The authors of [1] have explained that the failure of pFedMe [4] is attributed to two aspects that are irrelevant to data heterogeneity: i) the half-precision operator adversely impacts the performance and ii) the model can barely be updated due to the precision loss. Our method does not suffer from these two factors and, therefore, can effectively address data heterogeneity.
>
>
> [1] FederatedScope-LLM: A Comprehensive Package for Fine-tuning Large Language Models in Federated Learning. KDD 2024.
>
> [2] SPIDER: Searching Personalized Neural Architecture for Federated Learning. AAAI 2022.
>
> [3] Federated Fine-tuning of Large Language Models under Heterogeneous Tasks and Client Resources. NeurIPS 2024.
>
> [4] Personalized Federated Learning with Moreau Envelopes. NeurIPS 2020.

---

> > ### Comment · Reviewer_9rUg · 2024-11-23
> >
> > Thanks for the responses. After reading through your reply, I feel that my concerns remain unresolved.
> >
> >
> > ## Response to Weakness 4
> >
> > On one hand, even after quantization of the backbone, the proportion of bits occupied by LoRA remains negligible compared to the entire LLM. On the other hand, models that have been quantized usually need to be dequantized to a certain high precision for actual training and inference. After reading your reply, I highly doubt whether there truly exists a scenario where pruning only the LoRA parameters, which occupy an extremely small proportion, would enable an LLM unable to be loaded to just fit within the memory constraints of the current device. Even in such extreme cases, it seems there are alternative methods, such as reducing memory usage from the perspective of input length, which might be simpler and more practical than applying this technique.
> >
> >
> > ## Response to Question 2
> > In fact, I personally believe that this inconsistency in objectives is clearly evident. As discussed in [1], when applying LoRA to FL, there exists objective inconsistency between local training and global objective, due to the fact that the average of A multiplied by B is not equal to the product of the average of A and the average of B. Not to mention that this paper also prunes the weight matrices of LoRA. Therefore, I believe that simply claiming this inconsistency does not exist is unconvincing.
> >
> >
> > Overall, I believe the motivation of this manuscript needs further refinement. The current version does not sufficiently convince me of the importance of this work.
> >
> >
> > [1] Improving LoRA in Privacy-preserving Federated Learning. ICLR 2024.

---

> > > ### Author Response · Authors · 2024-11-24
> > >
> > > **Response to Weakness 4:**
> > >
> > > Thank you for your response. We want to clarify the feasibility and effectiveness of using LoRA-based fine-tuning for resources heterogeneous clients from the following three aspects:
> > > * Fine-tuning is essential for better performance [3]. The quantization methods on backbone LLMs are essential since they provide the prerequisite for fine-tuning and inference on local devices. The savings on resources and increases in token generation speed are proven to be effective [4-5].
> > > * A larger rank of LoRA leads to better performance [2-3]. Therefore, local clients should exploit their heterogeneous fine-tuning resources sufficiently.
> > > * The memory saving brought by alternative methods such as reducing input length is orthogonal to quantization and LoRA fine-tuning methods.
> > >
> > > **Response to Question 2:**
> > >
> > > We notice that [1] and [2] proposed different aggregation strategies for LoRA modules. [1] claims that the product of the averaged A and B involves additional terms that **may** neither benefit the optimization on the clients’ loss or FL global loss, while [2] disagrees with [1] and claims that the “reconstruction-distribution method” yields worse performance due to the lack of “cross-relation across clients. Therefore, the mechanisms behind aggregation methods for LoRA still remain unexplored.
> > >
> > > Our method proposes a new perspective of aggregation strategy identical to both [1] and [2] by applying element-wise aggregation on A and B on expanded sparse space.  The pruning process aims to identify personalized and global parameters by conducting element-wise NAS, rather than pruning weights during training.
> > > Together, [1], [2], and our method offer valuable research foundations in the area of FL-based LLM fine-tuning.
> > >
> > > [1] Improving LoRA in Privacy-preserving Federated Learning. ICLR 2024.
> > >
> > > [2] Heterogeneous LoRA for Federated Fine-tuning of On-Device Foundation Models. EMNLP 2024.
> > >
> > > [3] Federated Fine-tuning of Large Language Models under Heterogeneous Tasks and Client Resources. NeurIPS 2024.
> > >
> > > [4] AWQ: Activation-aware Weight Quantization for LLM Compression and Acceleration. MLSys 2024.
> > >
> > > [5] LLM.int8(): 8-bit Matrix Multiplication for Transformers at Scale. NeurIPS 2022.

---

> > > ### Author Response · Authors · 2024-11-30
> > >
> > > Dear Reviewer 9rUg,
> > >
> > > Sorry to bother you again.
> > >
> > > Given the importance of timely communication and as the rebuttal phase approaches its conclusion, we would like to confirm whether we have adequately addressed your concerns. If you have any remaining questions, please let us know. We are looking forward to your valuable reply.
> > >
> > > Thank you for your efforts in our paper.
> > >
> > > Sincerely,
> > >
> > > Authors

---

### Official Review · Reviewer_kdxA · 2024-11-01

**Soundness:** 2
**Presentation:** 2
**Contribution:** 2
**Rating:** 3
**Confidence:** 3

**Summary:**

The paper proposes a framework, Personalized Federated Instruction Tuning, which aims to enable personalized instruction tuning of large language models (LLMs) in federated settings. The approach incorporates Neural Architecture Search to allow each client to personalize their LoRA modules, thus addressing data and resource heterogeneity among clients. The framework includes personalized aggregation mechanisms to efficiently combine and redistribute updated parameters based on each client’s unique data and resource constraints.

**Strengths:**

- The paper addresses a meaningful problem of personalizing LoRA parameters for each client in a federated learning setup.
- Experimental results showcase PERFIT’s robustness across different LLMs, datasets, and client configurations
- By leveraging the pruning method to personalize and sparsely prune LoRA modules, the approach can minimize computational overhead and adapt to the computational capacities of different clients

**Weaknesses:**

- The description of the personalized aggregation module lacks clarity, particularly regarding the process of aggregating and redistributing mask and LoRA parameters across rounds. This complexity makes it challenging to fully grasp the module's function and purpose in the framework.
- The paper suffers from vague and inconsistent notation throughout, which makes it difficult for readers to follow the mathematical formulations and key concepts presented.
- The paper does not provide insights or analysis on adaptively setting the mask ratio for each client based on the data, which could be a significant parameter affecting performance based on individual client data distributions.
- Although the framework is positioned as utilizing NAS, the same base architecture is used across clients, with only varying degrees of unstructured pruning applied to LoRA modules, which may fall short of full architectural differentiation.
-  While a theorem is proposed, the paper does not provide detailed derivations, leaving gaps in the theoretical foundation and proof of the method’s performance. The insights behind the theorem should also be further explained. Why the $\kappa$ is negative? Please clearly explain the derivations of the theorem part.
- The evaluation relies solely on perplexity comparisons without examining time efficiency or computational costs, which are crucial for federated learning applications with resource constraints.
- The paper does not include baseline methods for comparison, which limits the ability to fully evaluate the effectiveness and rationale of the proposed approach. Including comparisons with simpler methods, such as fine-tuning the LoRA at each client as a personalization strategy, would provide valuable insight into its relative advantages and justify its complexity.

**Questions:**

Please refer to the weakness part.

---

> ### Author Response · Authors · 2024-11-22
>
> **Response to Weakness 1:**
>
> Sorry for the confusion. As shown in Figure 1, the personalized parameters of local clients are obtained by masking the global structure with personalized masks. Since the local fine-tuning is conducted on sparse matrices, the masks that align with the personalized parameters all have a value of 1. Note that our approach conducts the personalized aggregation element-wise. In other words, for each parameter whose mask value is 1, we aggregate it with other non-zero elements with the same index. We will add a figure to illustrate the personalized aggregation module in the final version.
>
> **Response to Weakness 2:**
>
> Thank you for pointing out the issues with notation in our paper. We will check the consistency of the notations and revise the paper.
>
>
> **Response to Weakness 3:**
>
> Thank you for your suggestions. In this paper, we mainly focus on conducting the element-wise architecture search in an expanded LoRA space instead of pruning for sparsity. Therefore, our method follows the design of NAS, which requires to pre-define the targeted resource. The targeted resource in our work is defined by the number of trainable parameters. Once the NAS process is finished, the searched architecture will be used for consecutive federated learning.
>
> **Response to Weakness 4:**
>
> Searching for the architectures of LLMs in federated learning is computationally expensive and unaffordable due to the limited resources of client devices. Therefore, we focus on element-wise neural architecture search on LoRA structures.  Due to the various sparse patterns our method can search, the diversity of the resulting architecture is significantly increased.
>
> **Response to Weakness 5:**
>
> Sorry for the confusion. Our theorem follows the derivations in [1], which provides the convergence rate for personalized federated learning with sparse local models. Since the training process on the searched parameter space equals the sparse local training on the expanded parameter space before applying NAS, the convergence analysis in [1] in the context of static masks is suitable for establishing the convergence rate for our method. Thank you for pointing out the mistake in $\kappa$, we will carefully check the derivations and provide the detailed steps following the work in [1] in the final version.
>
> **Response to Weakness 6:**
>
> Sorry for the confusion. We have already discussed the time efficiency and computational costs in lines 286-295.
>
>
> **Response to Weakness 7:**
>
> Thanks for your suggestions. We have conducted experiments based on the dataset, model and Rouge-L score used in [2] to show the advantages of our method. Note that the Rouge-L score is positively correlated with the human evaluation result, meaning that it is a valid indicator for evaluating the performance of LLMs [2].
> We keep the base rank of LoRA as 8. The rank and sparsity rate for our PerFIT is set to 12 and 33%, respectively. The rest hyperparameters follow the configuration defined in our PerFIT paper. The Rouge-L scores for standalone local fine-tuning, FIT and our PerFIT are 34.06, 55.15 and 57.05, showing that our method consistently achieves the best performance.
>
> [1] Achieving personalized federated learning with sparse local models. arXiv22.
>
> [2] Federated Fine-tuning of Large Language Models under Heterogeneous Tasks and Client Resources, NeurIPS 2024

---

> ### Author Response · Authors · 2024-11-30
>
> Dear Reviewer kdxA,
>
> Sorry to bother you again.
>
> Given the importance of timely communication and as the rebuttal phase approaches its conclusion, we would like to confirm whether we have adequately addressed your concerns. If you have any remaining questions, please let us know. We are looking forward to your valuable reply.
>
> Thank you for your efforts in our paper.
>
> Sincerely,
>
> Authors

---

### Official Review · Reviewer_UYo2 · 2024-11-05

**Soundness:** 2
**Presentation:** 2
**Contribution:** 2
**Rating:** 6
**Confidence:** 4

**Summary:**

Instruction tuning has been shown to be crucial for large language models in generating responses aligned with human preferences. This paper explores a novel method of collaborative instruction tuning under data privacy constraints and proposes a personalized federated instruction tuning framework (PerFIT). To address the data and resource heterogeneity among clients and prevent resource-rich clients from being limited by the constraints of resource-poor clients, the authors introduce an architecture auto-search method, allowing each client to obtain a personalized instruction tuning architecture. Overall, this paper provides an optimally configured model architecture for clients with heterogeneous resources.

**Strengths:**

S1: The paper introduces a federated instruction tuning framework based on architecture auto-search, effectively addressing data and resource heterogeneity in federated learning.
S2: The paper’s structure is well-organized, and the logic is clear.
S3: Ample theoretical analysis is provided, supporting the effectiveness of the proposed method.

**Weaknesses:**

W1: The discussion of federated instruction tuning for LLMs in related work is insufficiently in-depth, as it only briefly mentions two LoRA-based FIT frameworks that address data heterogeneity.
W2: In Figure 1, there are discrepancies between the legend (②, ③, and ④) and the explanations in the text. For example, in the text, ② represents “Sparse Module Generation and Local Fine-tuning,” but it appears as “Sparse Module Generation” in the legend. The fine-tuning process should be indicated on the specific sparse modules in the figure, possibly by adding an icon (e.g., a flame) to represent fine-tuning.
W3: The paper claims this method as the first solution to address the issue of personalized federated instruction tuning; therefore, experiments only compare it with the global model. However, some existing methods already address data heterogeneity in federated instruction tuning. It is suggested that the paper include a comparative analysis with these methods regarding data heterogeneity.
W4: The paper does not provide open-source code.

**Questions:**

Q1: The notation for i and j in the personalized aggregation section is confusing. In m, i and j represent client IDs, while in A, B, and I, they denote positions. A clearer notation is recommended.
Q2: The third step in the framework is “personalized model aggregation.” Based on the description in Algorithm 3, this personalization is implemented in a grouped aggregation manner. It would be more precise to refer to this as “grouped model aggregation” as “personalized aggregation” is somewhat coarse.
Q3: Equation (6) lacks a label (line 265).
Q4: The last sentence on page 5 and the first sentence on page 6 lack continuity; please check for any missing information.

---

> ### Author Response · Authors · 2024-11-22
>
> **Response to Weakness 1:**
>
> Thank you for your suggestion. We will include more discussions in related work on FIT methods including [1] and [2].
>
> **Response to Weakness 2:**
>
> Sorry for the confusion. We will modify the final version to correct the discrepancies.
>
> **Response to Weakness 3:**
>
> Thank you for your suggestions. We will add more analysis of existing personalized federated instruction tuning methods that address data heterogeneity, including [2].
>
> **Response to Weakness 4:**
>
> Sorry for the confusion. The code has been included in the supplementary material.
>
> **Response to Question 1:**
>
> Sorry for the confusion. We will carefully check the consistency of the notations.
>
> **Response to Question 2:**
>
> Thank you for your suggestion. We agree that it is more precise to use “grouped model aggregation” to explain the personalized parameter aggregation strategy. We will revise the paper in the final version.
>
> **Response to Question 3:**
>
> Thank you for pointing out the missing label for Equation 6. We will add the label in the final version.
>
> **Response to Question 4:**
>
> Thank you for pointing out the issue. We will carefully check and revise the unfinished sentence at the end of page 5 in our final version.
>
> [1] FederatedScope-LLM: A Comprehensive Package for Fine-tuning Large Language Models in Federated Learning. KDD 2024
> [2] FDLoRA: Personalized Federated Learning of Large Language Model via Dual LoRA Tuning. arXiv24.

---

> ### Author Response · Authors · 2024-11-30
>
> Dear Reviewer UYo2,
>
> Sorry to bother you again.
>
> Given the importance of timely communication and as the rebuttal phase approaches its conclusion, we would like to confirm whether we have adequately addressed your concerns. If you have any remaining questions, please let us know. We are looking forward to your valuable reply.
>
> Thank you for your efforts in our paper.
>
> Sincerely,
>
> Authors

---

### Comment · Area_Chair_nF2s · 2024-11-25

Dear reviewers,

As the deadline for discussion is ending soon. Please respond to the authors to indicate you have read their rebuttal. If you have more questions, now is the time to ask. This is important since the paper is currently undergoing extremely divergent scores.

AC

---

### Meta-Review · Area_Chair_nF2s · 2024-12-17

**Metareview:**

This paper proposes PerFIT, a personalized federated instruction tuning framework addressing data and resource heterogeneity in LLM fine-tuning. Despite attempting to tackle an important problem in federated learning, most of the reviewers recommend rejection due to significant methodological and experimental limitations. The consensus is that while the paper addresses a meaningful research challenge, its technical contributions are marginal, with most reviewers rating the soundness and contribution as "fair" (2 out of 5).

The primary weaknesses include: extremely limited performance improvements, lack of novel contributions, insufficient theoretical foundations, and unconvincing demonstrations of addressing resource heterogeneity (reviewer 9rUg). Critical concerns were raised about the method's claims, such as overclaiming the impact of parameter reduction when LoRA typically represents only 1% of model parameters. Additionally, reviewers noted methodological shortcomings like vague notation (kdxA), unrealistic experimental settings with IID data distribution (dUBm), and insufficient comparative analyses with existing federated learning approaches. The experimental evaluation was particularly criticized for comparing only with a single baseline and failing to provide comprehensive evidence of the proposed method's effectiveness across diverse scenarios.

**Additional Comments On Reviewer Discussion:**

No discussion. Reviewer yZsM mentioned that he/she have read others' comments and agreed this paper should be rejected.

---

### Decision · Program_Chairs · 2025-01-22

Reject